# Integrating spin-dependent emission and dielectric switching in Fe$^{II}$ catenated metal-organic frameworks

Xue-Ru Wu[1], Shu-Qi Wu[2]✉, Zhi-Kun Liu[3], Ming-Xing Chen[4], Jun Tao[3], Osamu Sato[2] & Hui-Zhong Kou[1]✉

Mechanically interlocked molecules (MIMs) including famous catenanes show switchable physical properties and attract continuous research interest due to their potential application in molecular devices. The advantages of using spin crossover (SCO) materials here are enormous, allowing for control through diverse stimuli and highly specific functions, and enabling the transfer of the internal dynamics of MIMs from solution to solid state, leading to macroscopic applications. Herein, we report the efficient self-assembly of catenated metal-organic frameworks (termed catena-MOFs) induced by stacking interactions, through the combination of rationally selected flexible and conjugated naphthalene diimide-based bis-pyridyl ligand (BPND), $[M^I(CN)_2]^-$ (M = Ag or Au) and Fe$^{2+}$ in a one-step strategy. The obtained bimetallic Hofmann-type SCO-MOFs $[Fe^{II}(BPND)\{Ag(CN)_2\}_2]\cdot3CHCl_3$ (**1Ag**) and $[Fe^{II}(BPND\{Au(CN)_2\}_2]\cdot2CHCl_3\cdot2H_2O$ (**1Au**) possess a unique three-dimensional (3D) catena-MOF constructed from the polycatenation of two-dimensional (2D) layers with **hxl** topology. Both complexes undergo thermal- and light-induced SCO. Significantly, abnormal increases in the maximum emission intensity and dielectric constant can be detected simultaneously with the switching of spin states. This research opens up SCO-actuated bistable MIMs that afford dual functionality of coupled fluorescence emission and dielectricity.

For more than half a century, mechanically interlocked molecules (MIMs) have aroused great research interest owing to their esthetic appeal and dynamic physical properties[1-6]. Introducing bistability into these MIM systems has proved to be of particular interest, which endows these systems with the capacity to switch between two distinct stable states when subjected to specific external stimuli, paving the way for various sophisticated molecular switches. Notable examples include rotaxanes and catenanes, which can exhibit switching behavior in response to changes in light, pH value, or redox states and are

expected to be applied to molecular electronic devices and drug delivery[7-11]. In recent years, a number of discrete interlocking molecular catenanes with increasingly intricate structures have been synthesized. In contrast, the design and synthesis of the [∞] catenated metal-organic frameworks (catena-MOFs) is still in its infancy[12-18]. The concept of robust dynamics envisages that MOF materials can significantly improve orderliness and performances while preventing degradation of their components during repeated switching processes[6,8], enabling the internal dynamics of MIMs to transfer from

[1]Engineering Research Center of Advanced Rare Earth Materials (Ministry of Education), Department of Chemistry, Tsinghua University, 100084 Beijing, PR China. [2]Institute for Materials Chemistry and Engineering & IRCCS, Kyushu University, 744 Motooka, Nishi-ku, Fukuoka 819-0395, Japan. [3]Key Laboratory of Cluster Science of Ministry of Education, School of Chemistry and Chemical Engineering, Liangxiang Campus, Beijing Institute of Technology, 102488 Beijing, PR China. [4]Analytical Instrumentation Center, College of Chemistry and Molecular Engineering, Peking University, 100871 Beijing, PR China. ✉e-mail: wu.shuqi.152@m.kyushu-u.ac.jp; kouhz@mail.tsinghua.edu.cn

solution to solid state, thus achieving macroscopic applications. One of the most critical issues for bistable catena-MOFs is how to promote switchable motion between two well-defined states, which can respond in situ to diverse stimuli[19,20].

To achieve these goals, it is essential to control the coupling between individual switchable molecules and the environment, as well as to integrate these bistable molecules into ordered components. Herein, we focus on molecular spin crossover (SCO) complexes of $3d^4$–$3d^7$ octahedral transition metal ions, whose spin states can be reversibly switched between high spin (HS) and low spin (LS) states, thus allowing the observation of the characteristic bistability in magnetism[21–26]. During spin transition, structural changes at the molecular level lead to crystal deformation, which can be used to promote a mechanical effect[27]. Given these structural changes, SCO complexes can serve as effective switching units. Any external stimuli capable of manipulating spin states, such as temperature, pressure, light irradiation, magnetic fields, and guest molecules, can be employed to operate the desired switching devices. This versatility leads to a broader range of switching techniques and widens the scope of potential applications[28]. Therefore, SCO molecules can be ideal candidates for developing micro- and macroscopic molecular switches for various physical properties, such as magnetism[29], conductivity[30,31], luminescence[32–37], dielectric properties[38–41], and mechanical effects[42,43]. Particularly, the spin-dependent synergistic switching of photoluminescence (PL) and dielectric properties, combined with the stiffness brought about by mechanical interlocking, makes SCO-based MIMs[44] promising for robust optoelectronic devices. This provides the opportunity for a bi-channel (i.e. optical and electrical) read-out of the magnetic states of such systems. Moreover, the influence of the dielectric background on photoluminescence, through adjusting the energy levels of excited states with different dipole moments and also the optical properties such as refractive index, presents a unique playground for understanding the interplay between these physical properties, providing different perspectives for the development of molecular switches and multi-channel devices.

The rational design of organic bis-pyridinyl linkers is the key to the construction of catena-MOFs. Flexible organic ligands facilitate the coexistence of coordination networks and p-stacked. In our work, the rigid naphthalene diimide (NDI) group is opted as the p-conjugated moiety[45], which contributes to favorable aromatic π-π stacking interactions and excellent luminescent properties. The sp³-hybridized carbon atoms within their flexible methylene group allow two pyridine arms to rotate freely, which in turn enables the supramolecular polymerization of the resulting catenanes via NDI-NDI dimerization[46–48]. We envisage that integrating the NDI-based ligand N, N'-bis(4-pyridylmethyl)–1,4,5,8-naphthalene diimide (BPND) with the classical bimetallic Hofmann-type $Fe^{II}\{M^I(CN)_2\}_2$ (M = Ag or Au) building blocks will lead to multifunctional catena-SCO MOFs.

In line with this strategy, we synthesized two catena-MOFs $[Fe^{II}(BPND)\{Ag(CN)_2\}_2]\cdot 3CHCl_3$ (**1Ag**) and $[Fe^{II}(BPND)\{Au(CN)_2\}_2]\cdot 2CHCl_3\cdot 2H_2O$ (**1Au**), which exhibited two-dimensional (2D) → three-dimensional (3D) parallel polycatenation. Both complexes undergo thermal- and light-induced spin transition. The reversible SCO was confirmed by variable-temperature single-crystal X-ray diffraction, magnetic susceptibility measurements, and Mössbauer spectroscopy. As expected, strong coupling between SCO and luminescence was clearly observed in **1Ag** and **1Au**. It is noteworthy that **1Ag** has two emission bands: the emission at around 465 nm is due to the monomer NDI groups, and the broad peak at around 625 nm at room temperature is attributed to intermolecular excimer from NDI groups. The intensities of both emissions are correlated with the spin state transition. Moreover, the dielectric constant of **1Ag** and **1Au** in the electronic configurations from LS to HS can be well observed in the temperature-dependent dielectric constant, which is coupled with the

SCO synchronously, confirming the SCO-induced dielectric switching behavior.

## Results

### Self-assembly and crystal structures

The reaction between BPND, $[M(CN)_2]^-$ (M = Ag and Au, respectively) and $Fe^{2+}$ in a molar ratio of 1:2:1 gave rise to two catena-MOFs **1Ag** and **1Au** (Fig. 1, Supplementary Figs. 1–3). Single crystals of **1Ag** and **1Au** were obtained by slowly liquid-to-liquid diffusing the methanolic solution of $Fe(ClO_4)_2\cdot 6H_2O$ into the chloroform-methanol solution of BPND and $[M(CN)_2]^-$. The homologous diamagnetic complex $[Zn^{II}(BPND)\{Ag(CN)_2\}_2]\cdot 3CHCl_3$ (**2Ag**) was similarly prepared (Supplementary Fig. 4). Variable-temperature single-crystal X-ray diffraction data were collected at 100, 170, and 250 K for **1Ag** and 100 K and 220 K for **1Au** to characterize the structural changes during spin transition. Supplementary Tables 1–5 summarize the crystal data, structure refinement parameters, and selected bond lengths for **1Ag, 1Au**, and **2Ag** at different temperatures.

Single-crystal X-ray diffraction analysis revealed that **1Ag** and **2Ag** crystallize in the polar tetragonal space group (Z = 16), $I4_1cd$ at all measured temperatures. While the Flack parameter for **1Ag** is close to 0.5, the polar crystal structure can be clearly verified by the photo-pyroelectric measurements: when the single crystals undergo instantaneous heating with a 532 nm laser, a transient electric current signal can be detected as a result of the change in the macroscopic polarization. This phenomenon can be repeatedly observed in the temperature range of 100–300 K, which falls outside the temperature range associated with the photomagnetic effect (Supplementary Fig. 5). Crystal data show that all $Fe^{II}$ ions are crystallographically equivalent. Each $Fe^{II}$ unit contains three chloroform molecules embedded within the hole of the framework (Supplementary Figs. 6 and 7). The presence of chloroform molecules can also be confirmed by infrared (IR) spectrum and elemental mapping photographs (Supplementary Figs. 8–10). The $Fe^{II}$ ions are bridged by four bidentate $[Ag(CN)_2]^-$ linkers in the equatorial plane to form rhombic $[Fe_4\{Ag(CN)_2\}_4]$ grids with sizes of 10.550(1) Å × 10.379(1) Å at 100 K (10.486(1) Å × 10.666(1) Å at 170 K, and 10.696(1) Å × 10.525(1) Å at 250 K) (Fig. 1b). The axial sites of the $[Fe^{II}N_6]$ octahedral are occupied by two nitrogen atoms of two BPND ligands. Two 4-pyridyl arms of the BPND ligand are in syn-conformation, connecting two $Fe^{II}$ ions on the diagonal in the rhombic grids (Fe···Fe distances are 13.473(3) Å and 13.335(3) Å at 100 K, 13.575(3) Å and 13.468(3) Å at 170 K, 13.665(2) Å and 13.547(2) Å at 250 K) (Fig. 1c). The resulting 6-connected structural units lead to the formation of a 2D layer with **hxl** topology (Supplementary Figs. 11 and 12)[49]. It is noteworthy that due to the bent conformation of BPND ligand, the layer thickness allows for the formation of entanglement by parallel polycatenation. Therefore, the adjacent 2D layers are interlocked by the bent BPND ligands, and the polycatenation of 2D layers gives rise to an overall 3D framework. Moreover, according to reports in the TopCryst database of ToposPro, among 2D layers of **hxl** topology no polycatenated examples have been found[16,49,50].

In each interlocked NDI unit, two NDI groups are parallel and arranged in an orthogonal array with the center-to-center and interplanar separations of 3.470(18) Å at 100 K (3.506(15) Å at 170 K and 3.504(13) Å at 250 K) (Fig. 1d), indicating the presence of strong π-π interactions. In addition, weak π-π stacking interactions exist between pyridine rings of ligands with the center-to-center distance of 3.580(3) Å and the dihedral angle between two pyridine rings is 1.1(6)° at 100 K (3.592(3) Å and 0.8(6)° at 170 K; 3.615(3) Å and 0.8(5)° at 250 K). Interestingly, the Cl···O distances between the solvent molecule chloroform and the NDI group are shorter than the sum of the van der Waals radii[51], i.e., C31-Cl2···O2A (100 K: 2.974(10) Å, 154.9(6)°; 170 K: 2.995(9) Å, 155.9(6)°; 250 K: 3.024(9) Å, 154.6(7)°) and C32-Cl6···O4B (100 K: 3.041(11) Å, 150.1(12)°; 170 K: 3.201(15) Å, 143.6(16)°; 250 K:

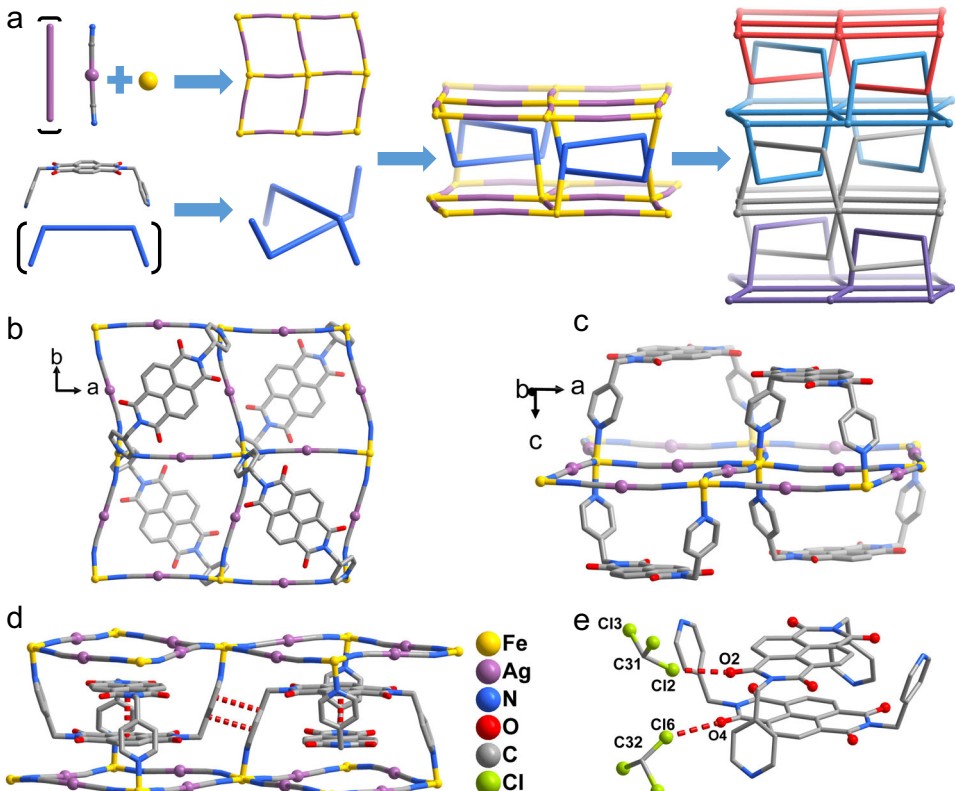

**Fig. 1 | Synthetic strategy and crystal structure of 1Ag. a** Self-Assembly of **1Ag** by the combination of BPND ligand, [Ag(CN)₂]⁻ and Fe^II ions. **b** and **c** Top and side view of the 2D layered structure for **1Ag. d, e** Supramolecular π-π stacking and C-Cl···O XB interactions in **1Ag** at 100 K. The red dashed lines represent supramolecular π-π stacking and C-Cl···O XB interactions. Hydrogen atoms and solvent molecules are omitted for clarity.

3.165(13) Å, 146.8(13)°), which indicates that there are two groups of halogen bond (XB, where X = Cl, Br, or I and B = nucleophile) non-covalent interactions (Fig. 1e)[52]. The formation of these supramolecular interactions builds a firm structure showing good thermal stability. The Fe-N bond lengths at 250 K are in the range of 2.106(6) −2.242(7) Å with an average value of 2.179(7) Å, indicating that Fe^II ions are in the HS state. The average Fe-N bond lengths are 2.156(8) Å (170 K) and 2.082(10) Å (100 K), corresponding to an incomplete and gradual interconversion from the HS to LS states of the Fe^II ions and match well with the magnetic susceptibilities (*vide post*).

The main 3D structure of **1Au** is basically isostructural to that of **1Ag**, in which the bridging ligand [Ag(CN)₂]⁻ was replaced with [Au(CN)₂]⁻ for **1Au**, with cell volume has been reduced by half (Supplementary Figs. 13–15). **1Au** crystallizes in the orthorhombic space group *Ccce* at 100 K and 220 K. Each Fe^II unit contains two chloroform molecules and two water molecules embedded within the hole of the framework (Supplementary Figs. 16 and 17). The oxygen atoms of water molecules and the chlorine atoms of chloroform molecules are crystallographically disordered in the framework at 100 K. The presence of chloroform molecules was confirmed by IR spectrum and elemental mapping photographs (Supplementary Figs. 18–20). Since the sp³-hybridized carbon atoms of methylene allow two pyridine arms to rotate freely, the asymmetric unit in **1Ag** contains two BPND ligands with different rotational angles (Supplementary Fig. 21), while **1Au** has only one type of BPND ligand. Therefore, different stacking modes between 2D layers are observed, the ABCD stacking pattern of **1Ag** and the ABAB stacking pattern of **1Au**, respectively (Fig. 2, Supplementary Figs. 15 and 22). For **1Au**, at 100 K, the π-π stacking interactions between the parallel pyridine rings have the interplanar distance of 3.298(18) Å and the center-to-center distance of 3.810(18) Å (3.397(10) Å and 3.778(10) Å, respectively, at 220 K), while 3.482(20) Å (3.505(13)

Å at 220 K) between parallel NDI centers in an orthogonal array (Supplementary Fig. 23). Moreover, at 100 K, the C5−H5···Cl5C (2.916(18) Å, 118.7(10)°) hydrogen bonds and the C17-Cl3···O1D (3.070(14) Å, 150.8(12)°) halogen bonds between the chloroform molecules and the framework lead to the removal of chloroform at an abnormally high temperature. The average Fe-N bond lengths are 2.178(8) Å (220 K) and 2.081(11) Å (100 K), corresponding to an incomplete spin transition in **1Au**.

## Magnetic properties and Mössbauer spectroscopy

The variable-temperature magnetic susceptibilities of **1Ag** and **1Au** on polycrystalline samples revealed that **1Ag** and **1Au** underwent gradual and incomplete spin transition without hysteresis (Fig. 3a, Supplementary Figs. 24 and 25). The $\chi_m T$ value of **1Ag** ($\chi_m$ is the molar magnetic susceptibility) is 4.08 cm³ K mol⁻¹ at 300 K, which is obviously larger than the spin-only value (3.0 cm³ K mol⁻¹) of a HS Fe^II ion. The potential electron transfer between the redox-active HS Fe^II and BPND should form HS Fe^III and reduced BPND ligand in **1Ag** with larger $\chi_m T$ values. However, the variable-temperature infrared absorption spectra for **1Ag** and its Zn^II analog **2Ag** show that the absorption features are similar and there are no bands for reduced BPND ligand, suggesting identical oxidation states for Fe and Zn ions in **1Ag** and **2Ag** (Supplementary Fig. 26). This result rules out the possibility of electron transfer in **1Ag**. The magnetic susceptibility was then calculated by the well-established ab initio protocol using the relativistic Complete-Active-Space Self-Consistent Field/N-Electron-Valence Perturbation Theory (CASSCF/NEVPT2) methods based on the truncated model extracted from the single-crystal structure of **1Ag** (250 K, HS, Supplementary Table 6). The results indicate that **1Ag** exhibits unquenched orbital angular momentum with a $\chi_m T$ value of 3.95 cm³ K mol⁻¹ at 300 K in this coordination geometry, consistent with the experimental

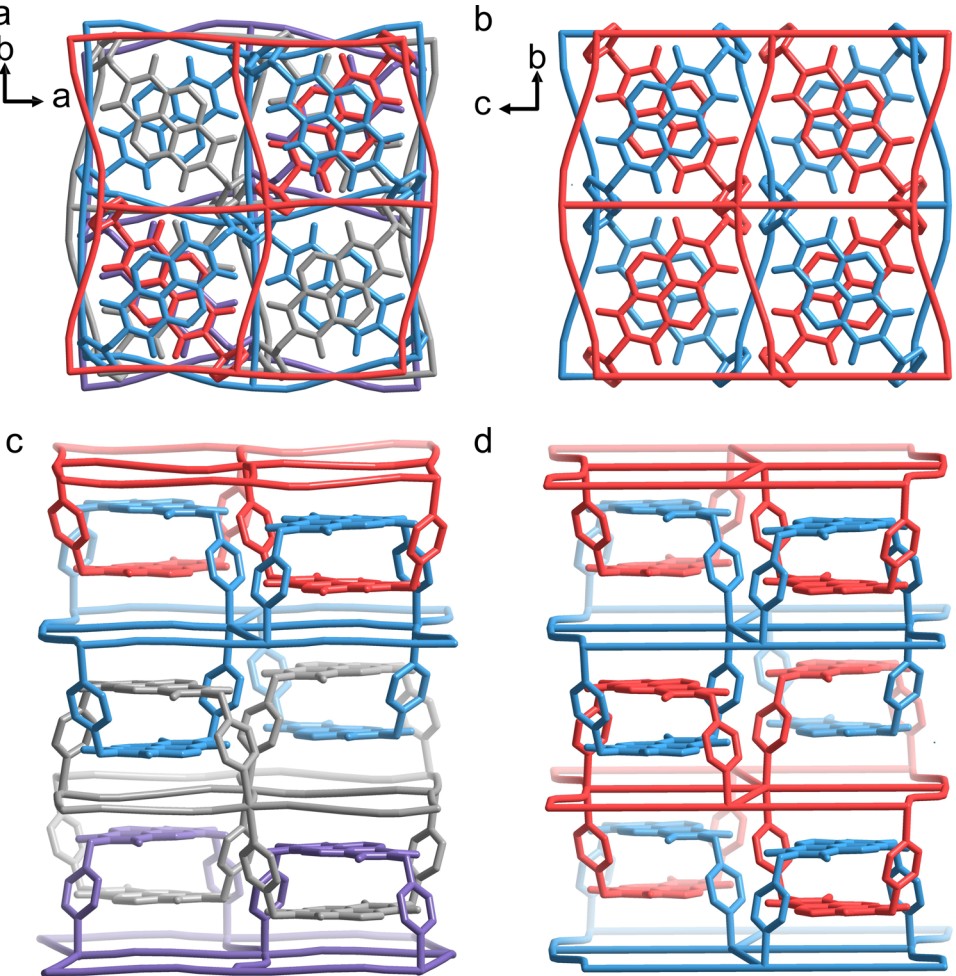

**Fig. 2 | Packing modes and entanglement of hxl 2D layers for 1Ag and 1Au.** **a** ABCD stacking pattern of 2D layers for **1Ag**. Four colors (red, blue, gray and purple) are used to illustrate interlocking adjacent 2D layers. **b** ABAB stacking pattern of 2D layers for **1Au**. **c**, **d** Illustration of parallel polycatenation in **1Ag** and **1Au**. Hydrogen atoms and solvent molecules are omitted for clarity.

values (Supplementary Fig. 27). Notably, the slight increase in the calculated $\chi_m T$ values upon cooling was also consistent with the experimental results (220–300 K). The $\chi_m T$ value remains almost constant upon cooling until 210 K and slowly decreases to 1.57 cm$^3$ K mol$^{-1}$ at 50 K. This value corresponds to ca. 38% of HS Fe$^{II}$ centers, indicating that 62% of Fe$^{II}$ centers changed from HS to LS state. A further decrease below 10 K is possibly due to zero-field splitting of the HS Fe$^{II}$ ions. The diffuse reflectance spectra display the absorption bands centered at 555 nm for **1Ag** (Supplementary Figs. 28 and 29). Therefore, irradiation to photo saturation was carried out using laser light of 532 nm (15 mW). When the sample was irradiated at 5 K for 4 h, the $\chi_m T$ value attained saturation at 2.45 cm$^3$ K mol$^{-1}$. After stopping irradiation, the $\chi_m T$ value increases up to the maximum value of 3.20 cm$^3$ K mol$^{-1}$ at 31 K. Complex **1Au** exhibits thermal- and light-induced SCO behavior similar to that of **1Ag** (Supplementary Fig. 30). At 300 and 50 K, the $\chi_m T$ values are 3.74 and 1.77 cm$^3$ K mol$^{-1}$, respectively, corresponding to about 53% of the Fe$^{II}$ ions undergoing spin transition.

To further verify the spin state of Fe$^{II}$ centers, the Mössbauer spectra (Fig. 3b) of **1Ag** were measured at room temperature and 50 K, respectively. The corresponding hyperfine parameters are summarized in Supplementary Table 7. At room temperature, **1Ag** has only one doublet with an isomer shift ($\delta$) of 1.061 mm s$^{-1}$ and a quadrupole splitting ($\Delta E_Q$) of 0.565 mm s$^{-1}$, indicating that all Fe$^{II}$ centers are in HS state ($S = 2$)[53]. At 50 K, one additional doublet corresponding to LS Fe$^{II}$

sites appears, indicating that partial Fe$^{II}$ centers (68.3%) have undergone the spin transition (LS Fe$^{II}$: $\delta = 0.499$ mm s$^{-1}$, $\Delta E_Q = 0.457$ mm s$^{-1}$; HS Fe$^{II}$: $\delta = 1.147$ mm s$^{-1}$, $\Delta E_Q = 0.999$ mm s$^{-1}$). It is worth noticing that the $\Delta E_Q$ values for HS Fe$^{II}$ are somewhat smaller than that previously reported (1.43–2.95 mm s$^{-1}$)[54], probably due to the less distorted local surrounding of HS Fe$^{II}$ in **1Ag** that causes a small deviation of the iron nucleus from the sphere. Unfortunately, it is difficult to observe signals for **1Au** at room temperature and low temperature (50 K), which might be related to the presence of heavy Au$^{I}$ ions in the complex that absorb the incident γ-radiation, hindering the resonance absorption of Fe$^{II}$[55].

### Dielectric properties

For HS octahedral Fe$^{II}$ complexes, two electrons occupy anti-bonding 3d orbitals ($e_g$), resulting in the increase of coordination bond lengths and structural distortion with the spin transition. Consequently, changes in local electrical dipoles caused by spin transition can be expected, leading to changes in dielectric properties during spin transition[41]. The variation of complex dielectric permittivity $\varepsilon^*$ ($\varepsilon^* = \varepsilon' - i\varepsilon''$, where $\varepsilon'$ and $\varepsilon''$ are the real and imaginary parts of $\varepsilon^*$) was measured between 15 to 300 K at different electric field frequencies for **1Ag** and **1Au** (Fig. 4, Supplementary Fig. 31). The frequency scans were carried out isothermally. The temperature-dependent dielectric constant $\varepsilon'$ at 10$^5$ Hz (in the heating mode) is compared with the magnetic susceptibility data (Fig. 4a). The dielectric constant is 2.90 at 50 K, and then gradually increases until 245 K reaches the

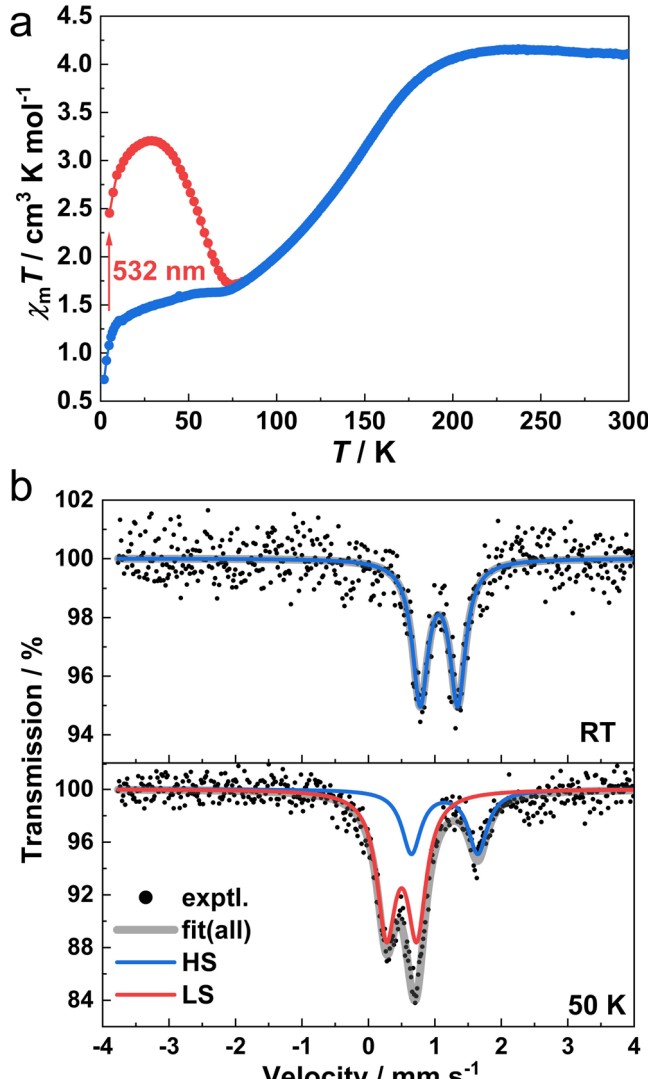

**Fig. 3 | Magnetic properties and $^{57}$Fe Mössbauer spectra of 1Ag. a** Temperature-dependent $\chi_m T$ product and light-induced spin transition effect induced by laser. The measurements were performed with a sweeping rate of 2 K min$^{-1}$. Temperature-dependent $\chi_m T$ product before (blue) and after irradiation (red) with 532 nm laser light in the heating mode. **b** $^{57}$Fe Mössbauer spectra at room temperature and 50 K, respectively.

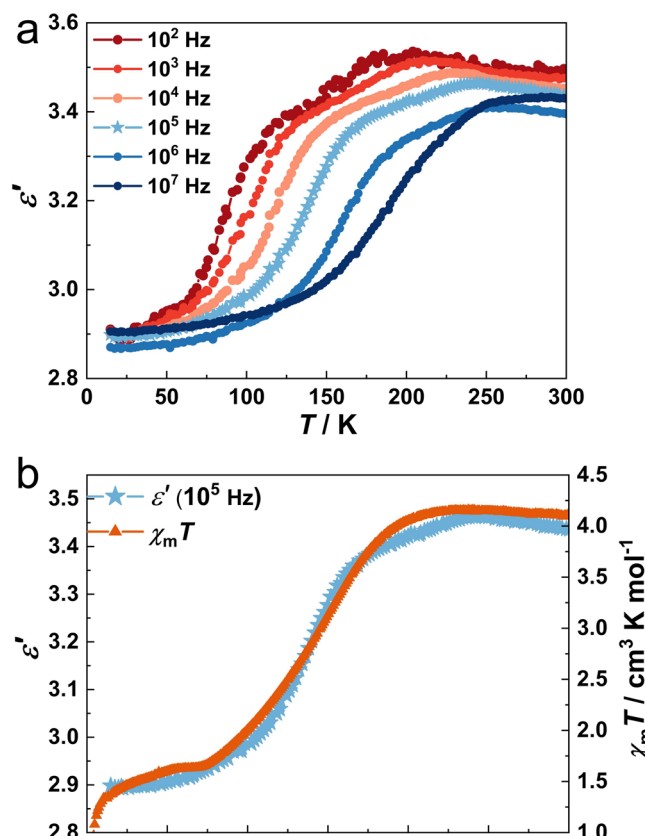

**Fig. 4 | Dielectric properties between 15 and 300 K of 1Ag. a** Temperature-dependent dielectric constants ($\varepsilon'$) versus frequency plots ($f = 10^2$ Hz to $10^7$ Hz) with a sweeping rate of 2 K min$^{-1}$. **b** $\varepsilon'$ ($f = 10^5$ Hz) as comparison with the $\chi_m T$ products in the heating mode.

Theoretically, the octahedral [FeN$_6$] geometry of Fe$^{II}$ ions is sensitive to its spin state, and the distorted octahedron stabilizes the HS state owing to the Jahn-Teller effect for its $t_{2g}^4 e_g^2$ electronic configuration. To verify this, the variation has been quantified by calculating the octahedral distortion parameter $\Sigma$, which is defined by the sum of the deviations from 90° of the 12 *cis*-N-Fe-N angles[58]. The $\Sigma$ value increased from 31.9° at 100 K to 36.3° at 250 K (Supplementary Table 8). The local symmetry distortion changes the local electric field significantly, and large changes in the dipole moment can be detected by the dielectric spectrum.

The dielectric constant $\varepsilon'$ and dielectric loss tan $\delta$ of **1Au** showed a temperature-dependent variation trend similar to that of **1Ag** (Supplementary Fig. 32 and Table 9). To elucidate the correlation between spin crossover and dielectric properties, the dielectric constants of the homologous diamagnetic Zn$^{II}$ complex **2Ag** at variable temperatures were measured under the same conditions (Supplementary Fig. 33). The dielectric constant $\varepsilon'$ and dielectric loss tan $\delta$ of **2Ag** remained almost unchanged from 15 to 300 K. Therefore, the spin transition of Fe$^{II}$ ions should be responsible for the dielectric switching response.

**Photoluminescence properties**

In order to verify the correlation between SCO and photoluminescence, the temperature-dependent emission spectra for pure BPND ligand, **1Ag, 1Au** and **2Ag** were measured in the heating mode. The emission intensity of free BPND ligand decreases monotonically with increasing temperature due to the expected thermal quenching effect (Supplementary Fig. 34). The excitation wavelength of 355 nm

maximum value of 3.46, which is consistent with the gradual SCO behavior of **1Ag**. During the SCO process, frequency-dependent peaks in dielectric loss (tan $\delta$, tan $\delta = \varepsilon''/\varepsilon'$) and imaginary part of the dielectric constant ($\varepsilon''$) were also observed, characteristic of dielectric relaxation (Supplementary Fig. 31). Notably, this phenomenon has been observed in other reported SCO systems[56,57]. The corresponding temperature-dependent relaxation time could be well fitted by the Arrhenius law, $\tau = \tau_0 \exp(E_a/k_B T)$ (where $\tau_0$ is defined as the pre-exponential factor, $E_a$ is the activation energy, $T$ is the temperature of the $\varepsilon''$ peak, and $k_B$ is the Boltzmann constant), giving the activation energy $E_a$ of 18.0(6) kJ/mol and $\tau_0$ of $2.6 \times 10^{-13}$ s (Supplementary Fig. 31c). This low activation energy, along with the broad temperature range of the peak shift, suggests that the SCO transition likely occurs in a non-correlated manner across a wide timescale. Such findings are in agreement with the gradual SCO behavior observed through magnetometry. These results directly demonstrate that the spin state-dependent dynamic local electrical dipoles change in **1Ag**.

was determined based on the excitation spectra of **1Ag** and **1Au** (Supplementary Figs. 35 and 36).

At 80 K, the emission spectrum of **1Ag** shows a dual emission peak: one is due to the monomer species of NDI groups at around 465 nm ($\lambda_1$), and the other is an unstructured broad peak at around 640 nm ($\lambda_2$), which is assigned to the intermolecular excimer emission (Fig. 5a, Supplementary Figs. 37 and 38). As temperature increases, the intensity of monomer emission ($\lambda_1$) increases slowly to 120 K, then increases rapidly, reaches the maximum at 210 K, and then decreases due to the thermal quenching effect. The emission intensity of the excimer ($\lambda_2$) decreases rapidly from 80 K to 120 K, and then slowly to 170 K. Above 170 K, the emission intensity begins to increase and reaches the maximum value at 210 K (Fig. 5b). The abnormal and discontinuous change of emission intensity of **1Ag** is obviously different from the monotonic decrease of the pure ligand (Supplementary Fig. 39). **1Ag** exhibits gradual SCO in a relatively wide temperature range of 50–210 K. The abnormal temperature range of monomer emission falls within the range of spin transition temperature, certificating that the intensity of luminescence is mainly controlled by the spin state of $Fe^{II}$ ions. Similarly, the inflection point of excimer emission intensity appears at 210 K, but its intensity change is not as obvious as that of the monomer. The lattice expansion during the spin transition process ($\Delta V/V \sim 4\%$) may induce spatial distortion in the formation of NDI-NDI excited dimerization. Such mechanically induced interference leads to changes in the emission displacement and intensity of excimers[59,60]. In our previous studies, the luminescence-SCO coupling originated from the energy transfer between the luminescence donor and the spin center receptor[61]. Therefore, the regulation of spin state

transition on excimer emission is reflected in the change of spectral overlap and interplanar NDI-NDI dimerization distance.

Unlike **1Ag**, the emission spectrum of **1Au** only contains an emission peak of around 460 nm, which is attributed to the monomer species of NDI (Supplementary Fig. 40). Considering that the NDI-NDI stacking in **1Au** is very similar to that in **1Ag**, it is unexpected that no excimer emission was observed in **1Au**, which may be related to the existence of some non-radiative transition pathways. The monomer emission intensity of **1Au** shows a temperature-dependent variation trend similar to that of **1Ag**, while the inflection point appears at 220 K (Supplementary Figs. 41 and 42). These results clearly show that both complexes have an apparent coupling effect between luminescence and SCO properties. To further elucidate the regulation of the spin state transition of $Fe^{II}$ ions on the emission properties, we measured the temperature-dependent emission spectra of the $Zn^{II}$ complex **2Ag**. The emission intensity monotonically decreases with increasing temperature (Supplementary Figs. 43 and 44) owing to the thermal quenching effect.

The UV-vis diffuse reflectance spectra of **1Ag** and **1Au** at different temperatures (Supplementary Figs. 28 and 29) were recorded to better elucidate the coupling mechanism between SCO and luminescence. For **1Ag** at 80 K, there are two well-separated absorption bands at 400 nm and 550 nm, respectively. As the temperature increases, the absorption intensity gradually decreases. Similar temperature-dependent absorption features can also be found for **1Au**. Therefore, an energy transfer mechanism should be responsible for the SCO-luminescence coupling effect.

The time-dependent density functional theory (TD-DFT) calculations were performed to investigate the vertical excitation energies and the corresponding oscillation strengths for both the HS and the LS structures of **1Ag** at B3LYP/def2-SVP level (Supplementary Figs. 45 and 46). Supplementary Table 10 summarizes selected excitation energies, corresponding oscillator strengths ($f$) and the assignments of absorption peaks for LS **1Ag** and HS **1Ag**. As shown in Fig. 6, the two absorption bands of LS **1Ag** centered at 466 nm and 548 nm overlap the emission band effectively. Therefore, luminescence is quenched through energy transfer. The two absorption bands of HS **1Ag** are blue shifted to 420 nm and 518 nm respectively, far away from the emission band, and the oscillator strength is markedly lower than that of LS state. Consequently, it was observed that the emission intensity synchronously increases with the increase of the proportion of HS $Fe^{II}$ ions.

In conclusion, we have successfully synthesized and characterized two $Fe^{II}$ SCO-based catenated MOFs. Both complexes exhibit thermal- and light-induced spin transition, verified by temperature-dependent

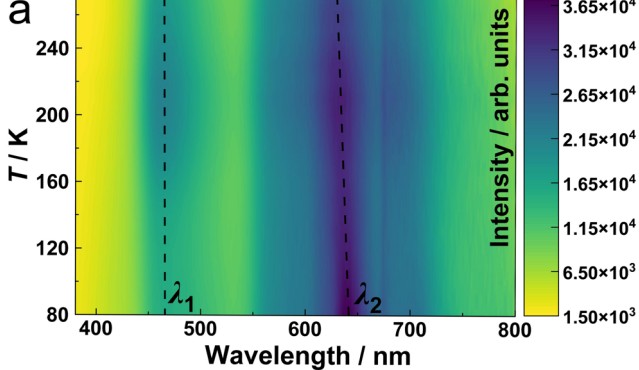

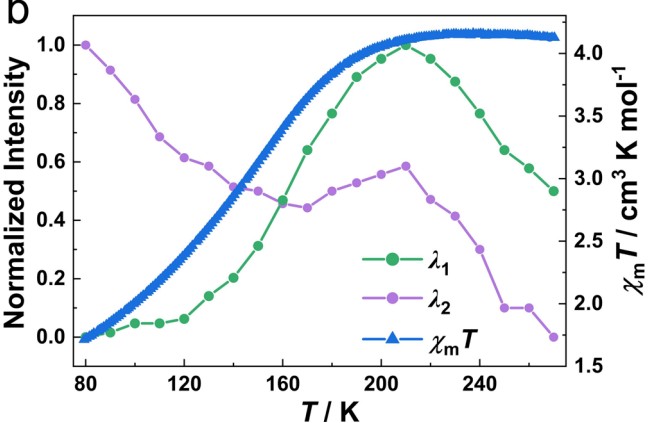

**Fig. 5 | Luminescent properties of 1Ag. a** A 2D color map of the temperature-dependent emission spectra ($\lambda_{ex} = 355$ nm). Dashed lines represent the characteristic emission peaks of monomer ($\lambda_1$) and excimer ($\lambda_2$). **b** Normalized maximum emission intensity ($\lambda_1$ and $\lambda_2$) as comparison with the $\chi_m T$ products in the heating mode.

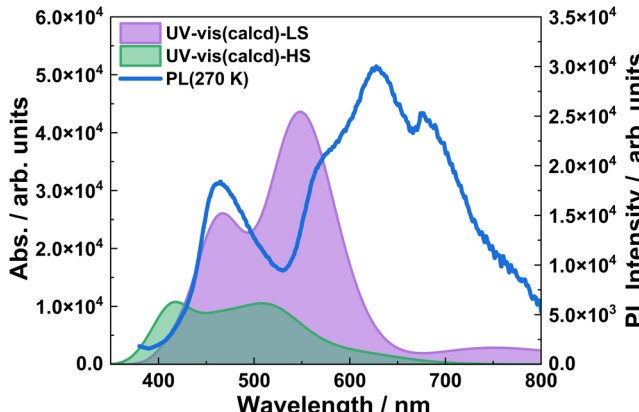

**Fig. 6 | TD-DFT-calculated absorption spectra of 1Ag.** Electronic absorption spectra predicted by TD-DFT for LS and HS **1Ag** and the photoluminescence emission spectrum at 270 K. The full width of the half-maximum is set to be 0.3 eV.

Mössbauer spectra, structural analyses, and magnetic measurements. Moreover, temperature-dependent emission spectra and dielectric constant demonstrated that the dual coupling of emission and dielectric properties were regulated by the spin state, where the coupling of SCO-luminescence/dielectricity comes from the energy transfer caused by spectral overlap (LS state), or the change of local electrical dipoles caused by structural deformation, respectively. These results provide strong evidence for the practical application of SCO materials. The mechanically interlocked structure improves the stiffness of SCO materials and is conducive to enhancing the coupling effect between luminescence and SCO. Correspondingly, the introduction of SCO units brings magnetic, optical, and electrical properties to bistable MIM materials, which makes SCO-based MIMs promising for advanced sensing materials. Research to explore these possibilities is underway in our laboratory.

## Methods

### Synthesis
All chemicals and solvents were purchased from commercial sources and used without further purification.

### N,N′-bis(4-pyridylmethyl)-1,4,5,8-naphthalene diimide (BPND)
BPND was synthesized by adapting literature methods[62]. 1,4,5,8-Naphthalenetetracarboxylic dianhydride (4.5 mmol) and 4-(aminomethyl)pyridine (11.3 mmol) were refluxed in anhydrous DMF (25 mL) under $N_2$ atmosphere 12 h, an orange-brown suspension was obtained. The solid was isolated by vacuum filtration and washed with dichloromethane and acetone before vacuum drying. Yield: ~85%. Anal. Calcd for $C_{26}H_{16}N_4O_4$: C, 69.64; H, 3.60; N, 12.49. Found: C, 69.56; H, 3.65; N, 12.54.

### [Fe$^{II}$(BPND){Ag(CN)$_2$}$_2$]·3CHCl$_3$ (1Ag)
A mixture of ligand (0.02 mmol) in chloroform (3 mL) and Na[Ag(CN)$_2$] (0.04 mmol) in methanol (1 mL) was placed in the bottom of a test tube, and then a mixture of methanol and chloroform [1:1 (v/v); 3 mL] was added as a buffering solution. A methanol solution (0.5 mL) of Fe(ClO$_4$)$_2$·6H$_2$O (0.02 mmol) was carefully added to the top of the test tube. The test tube was sealed and left undisturbed at room temperature. Brown crystals were obtained after 2 weeks in the second layer (yield: ~50%). Anal. Calcd. for $C_{33}H_{19}Ag_2Cl_9FeN_8O_4$: C, 33.53; H, 1.62; N, 9.48. Found: C, 33.46; H, 1.69; N, 9.43.

### [Fe$^{II}$(BPND){Au(CN)$_2$}$_2$]·2CHCl$_3$·2H$_2$O (1Au)
**1Au** was synthesized by following the method of **1Ag** except that Na[Ag(CN)$_2$] and Fe(ClO$_4$)$_2$·6H$_2$O were replaced with K[Au(CN)$_2$] and FeCl$_2$·4H$_2$O, respectively. (yield: ~40%). Anal. Calcd. for $C_{32}H_{22}Au_2Cl_6FeN_8O_6$: C, 30.10; H, 1.74; N, 8.77. Found: C, 30.22; H, 1.70; N, 8.70.

### [Zn$^{II}$(BPND){Ag(CN)$_2$}$_2$]·3CHCl$_3$ (2Ag)
**2Ag** was synthesized by following the method of **1Ag** except that Fe(ClO$_4$)$_2$ was replaced with Zn(ClO$_4$)$_2$ (yield: ~40%). Anal. Calcd. for $C_{33}H_{19}Ag_2Cl_9ZnN_8O_4$: C, 33.26; H, 1.61; N, 9.40. Found: C, 33.32; H, 1.55; N, 9.39.

### Single-crystal X-ray diffraction
Single crystal X-ray diffraction data for **1Ag** and **2Ag** were collected using a Rigaku SuperNova, Dual, AtlasS2 diffractometer. Single-crystal X-ray diffraction data for **1Au** on XtaLAB Synergy R diffractometer using Rigaku (Cu) X-ray Source (Rotating-anode X-ray tube). The structures were solved by direct method Olex2 1.3 and refined by full-matrix least-squares (SHELXL or Olex2 1.3) on $F^2$. Hydrogen atoms were added geometrically and refined using a riding model. For **1Au** (220 K), SQUEEZE was employed to calculate the diffraction from the solvent region and thereby produced a set of solvent-free diffraction intensities. A total of 1172 electrons per unit cell were masked, consistent with 16 chloroform molecules and 16 water molecules (two chloroform molecules and two water molecules per Fe$^{II}$ unit).

### Magnetic measurements
Variable-temperature magnetic susceptibility measurements were carried out on a Quantum Design MPMS XL7 magnetometer under a magnetic field of 5000 Oe at a temperature range of 2–300 K with a sweeping rate of 2 K min$^{-1}$. All data were corrected for diamagnetic contributions[63]. Photomagnetic measurements were performed on the powdered sample attached to transparent tape. A green laser (532 nm) was adopted as the excitation source.

### Fluorescence spectroscopy
Emission spectra of fluorescence were measured on an Edinburgh FLS 980 fluorescence spectrophotometer. The samples were placed in an electrically heatable continuous flow cryostat (Oxford Instruments). The waiting time for each temperature stabilization was 300 s.

### Electrical property measurements
Temperature-dependent dielectric constants were measured using the two-probe *a.c.* impedance methods in the frequency range from 100 Hz to 10 MHz on a Wayne Kerr 6500B Precise Impendance Analyzer. The electric contacts were prepared by using silver paste (DAD-87) to attach 25 μm gold wires to the pressed powder samples. The sample was placed into a Janis cryogenic refrigeration system with a warming rate of 2 K min$^{-1}$. Pyroelectric measurements were performed with Keithley 6517B electrometer and the Quantum Design MPMS-XL chamber as temperature controller. The single-crystal sample (0.2 × 0.18 × 0.05 mm$^3$) was sandwiched between the silver pastes on its (001) and (00−1) surfaces.

### Other physical measurements
Elemental analyses (C, H, and N) were performed on a Cario Erballo elemental analyzer. The UV-vis reflectance spectra were recorded on a Shimadzu UV 3600 UV-vis-NIR spectrophotometer under $N_2$ atmosphere. Thermogravimetric analyses (TGA) were performed on a DTU-3A simultaneous thermal analyzer from room temperature to 800 °C under $N_2$ atmosphere at a heating rate of 5 °C min$^{-1}$. $^{57}$Fe Mössbauer spectra were recorded on a Topologic 500AV spectrometer. Powder X-ray diffraction (PXRD) data were measured by Cu K$\alpha$ radiation ($\lambda = 1.5418$ Å) on a Rigaku diffractometer with a scanning rate of 5° min$^{-1}$ at 298 K. The infrared (IR) spectra were recorded on a WQF-510A FTIR spectrometer. Field-emission scanning electron microscopy (FE-SEM) images were performed on a SU-8010 scanning electron microscope.

### Density functional theory calculations
The ab initio calculations were performed with the ORCA program package version 5.0.3[64]. The CASSCF method was used to take the static correlation effect into account, while the NEVPT2 correction was adopted to consider the dynamic correlation effect. The def2-TZVP[65] basis set was used for all atoms. The magnetic susceptibility was calculated in a 5000 Oe external field.

Truncated molecular models (containing four Fe$^{II}$ ions, each Fe center coordinated by one BPND ligand and one pyridine molecule in the axial direction, and four CN$^-$ on the equatorial plane) were constructed based on the single-crystal structure of **1Ag** measured at 100 K and 250 K without further optimizations for the TD-DFT calculations (Supplementary Data 1). The TD-DFT calculations were performed in ORCA 5.0.3 program, and the Tamm Dancoff Approximation (TDA) approximation method was used to investigate the vertical excitation energies and the corresponding oscillation strengths on the models at B3LYP/def2-SVP level[65,66]. Simultaneously using auxiliary fitting basis sets def2-SVP/C[67] and def2/J[68] to support the RIJCOSX

density fitting approximation method. To cover the energy range of interest, 360 excited states were included in the TD-DFT calculations for both HS and LS states. The assignments of absorption peaks were made based on the CI expansions and charge density difference calculated by Multiwfn 3.7[69]. The molecular orbital diagrams were made by ChemCraft (Chemcraft - graphical software for visualization of quantum chemistry computations. https://www.chemcraftprog.com).

## Data availability

Crystallographic data for the structures reported in this article have been deposited at the Cambridge Crystallographic Data Centre under deposition numbers CCDC 2249939-2249941, 2249942-2249943 and 2269751. Copies of the data can be obtained free of charge via www.ccdc.cam.ac.uk/structures/. All other relevant data generated and analyzed during this study, which include spectroscopic, crystallographic and TD-DFT data, are included in this article and its supplementary information. The source data underlying Figs. 3–6, Supplementary Figs. 2–6, 8, 16, 18, 24–44 are provided as a Source Data file. Source data are provided with this paper. Figshare dataset: https://doi.org/10.6084/m9.figshare.25058381 (2024). Source data are provided with this paper.

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

## Acknowledgements

This work was supported by the National Natural Science Foundation of China (Grant Numbers 22271171 (H.-Z.K.), 21971142 (H.-Z.K.), 22371015 (J.T.)) and JSPS KAKENHI (Grant Numbers 24K17698 (S.-Q.W.), 24H00466 (O.S.)). We thank the Tsinghua Xuetang Talents Program for providing instrumentation and computational resources.

## Author contributions

X.-R.W., S.-Q.W. and H.-Z.K. designed the study and wrote the manuscript. X.-R.W. synthesized the materials and performed most of the experimental measurements. S.-Q.W., Z.-K.L. and J.T. contributed to magnetic measurements. M.-X.C. contributed to fluorescence spectroscopy. X.-R.W. and S.-Q.W. conducted theoretical calculations. S.-Q.W. and O.S. contributed to pyroelectric measurements. S.-Q.W. and H.-Z.K. supervised the study. All authors discussed the results and commented on the manuscript.

## Competing interests

The authors declare no competing interests.
