## [Peer Review File · Nature Communications]

Integrating spin-dependent emission and dielectric switching in Fell catenated metal-organic frameworksREVIEWER COMMENTS

Reviewer #1 (Remarks to the Author):

The authors reported two interesting catenated metal-organic frameworks in the spin-crossover field. The spin-dependent synergistic switching of dielectricity and fluorescence emission were reported. The diamagnetic zinc complex was also synthesized to compare the spin-crossover iron complex to illustrate the spin-dependent dielectricity and fluorescence properties. The time-dependent density functional theory calculations were further performed to illustrate the absorption spectra. These complexes were well characterized. However, some problems should be fixed before publication.

1. The authors should check the address for all authors. For example, Prof. Jun Tao should belong to Beijing Institute of Technology.
2. Line 116 and 117: "Each FeII unit contains three chloroform molecules embedded within the hole of the framework, which is consistent with thermogravimetric analyses (TGA) (supplementary Figs. 5 and 6)." The chloroform has a low boiling point of 61.2°. However, the TG data show complex 1Ag can be stabilized to ~150°. It is possible that the chloroform molecules had already escaped from the sample before the TG measurement. Similarly, the TG of 1Au in Figure S11 also has the same problem. Complex 1Au can be stabilized to ~175°, which excludes the presence of methanol molecules.
3. The χ_{MT} value of 1Ag is 4.48 cm³ K mol⁻¹, which is out of range for a HS Fe(II) ion. It is possible that complex 1Ag contains the paramagnetic impurity. The author should carefully check the sample and make sure of the purity. Moreover, the solvents escaped from the sample as shown in TG figure. How about the sample for the magnetic measurement in Figure 3 and Figure S20. The authors should check the magnetic data with and without solvent molecules for 1Ag and 1Au.
4. Is the solvent still in the sample under the condition of photomagnetic measurements?
5. The assignment of HS and LS in Figure 3 is wrong. The isomer shift in Table S6 looks like inaccurate when comparing with Figure 3. The author should check the Mössbauer spectroscopic parameters.
6. The solvent problem should be also checked for the dielectric and fluorescence measurements.
7. The dielectric properties are extremely sensitive to the frequency. Such the dramatic frequency dependence is very weird. The authors should explain it.
8. In Line 247 and 248, the author stated that the monotonic decrease of the pure ligand (Figure S24). However, only fluorescence spectra were shown in Figure S24. The temperature dependent emission intensities at 465 and 640 nm should be provided for the pure ligand.
9. The temperature dependent emission intensities at 460 should be provided for the pure ligand to compare with that of 1Au.
10. The cartesian coordinate information should be provided as a supporting information for the truncated molecular models.

Reviewer #2 (Remarks to the Author):

Mechanically interlocked molecules (MIMs), such as catenanes, exhibit switchable properties and are of interest as components in a variety of stimuli responsive materials. In this submitted manuscript, Wu et al. report in detail the synthesis, crystal structures, and functions of novel bimetallic Hoffman-type MOFs based on a bis-pyridyl naphthalene diimide (BPND). These MOFs have an interesting three-dimensional structure supported by a catenane-type structure formed by BPND ligands, which is found to undergo SCO upon heat or light. Furthermore, anomalous increases in maximum luminescence intensity and dielectric constant were detected simultaneously with the spin-state switching. Although these interesting dual-functional materials have been demonstrated, the authors would need to further discuss the following points.

(1) What is the significance of combining spin-dependent emission and dielectric switching in a single material? Are there any advantages for practical applications?

(2) Correlations between catenated structures and functions

Is it possible to discuss the effect of catenane structures on physical properties more

straightforwardly by comparing with similar compounds that do not have catenated structures? For example, is it possible to synthesize MOFs in which the BPNDs are not catenated by changing the synthesis conditions?

Minor point

(1) Chemical structure of the ligand with naphthalene diimide core

The chemical structure of BPND (bispyridyl naphthalene diimide) drawn by software such as ChemDraw is not listed anywhere, including the main text and the Supplementary Information. I recommend that such a chemical structure be included in the appropriate place for the better understanding of the readers.

Reviewer #3 (Remarks to the Author):

The manuscript by Wu et al. reports on the synthesis and structural characterization of polycatenated bimetallic Fe(II)/Ag(I) and Fe(II)/Au(I) coordination networks of the Hofmann-type. The thermal and light induced SCO behaviour is also investigated. The reported materials show also spin-dependent photoluminescence and dielectric switching.

The novelty claimed by the Authors concerns the development of coordination networks showing the feature of mechanical link through catenation to be exploited in the integration of different spin-dependent functionalities. Although materials of the Hofman-type showing thermal and light-induced SCO as well as spin-dependent dielectricity and photoluminescence have been already reported in the literature, the novelty here is the achievement of an ordered organization of the chromophores in the solid state through catenation and pi-pi interactions. In general, the use of mechanical links in the solid state to control macroscopic properties is very interesting, however, in this manuscript the effect of catenation on the studied properties is not well investigated and described.

The structures are described as catena-MOF in which interpenetration occurs between 2D layers. However, the entanglement shown by the present structures is better described as polycatenation instead of interpenetration. The difference relates on whether there is or not an increase of dimensionality of the entangled array: in polycatenation there is such an increase, e.g. from 2D to 3D, in interpenetration the dimensionality of the single coordination motif is preserved e.g. from 2D to 2D (see ref. 17 of this manuscript and Carlucci et al. *Coord. Chem. Rev.* 2003, 246, 247). The analysis of the entanglement in the structures by the program ToposPro (Blatov et al. *Cryst. Growth Des.* 2014, 14, 3576) reveal that the layers have the 6-connected hxl topology. Moreover, an interesting aspect overlooked by the Authors is that among 2D layers of hxl topology no polycatenated examples have been found according to what reported in TopoCryst database of ToposPro. These aspects related to the topology and entanglement of the reported structures could be added to the manuscript.

A point of weakness of the manuscript concerns the structural analysis. Geometrical parameters such as e.g. Fe-N distances, pi-pi stacking distances and N-Fe-N angles are given to support spin-transition at different temperatures. However, no standard deviations for both single and average values are reported and, given the quality of the crystal structures, differences appear to be not very significant.

There are problems with the structure of 1Au at 220 K. There are Alerts of type A in the CheckCIF file concerning the refinement of the atomic displacement parameters (ADP) of the gold atoms. The apparent impossibility to refine a reasonable ADP for such a heavy atom cast doubts on the proposed structural model. In any case, the description of the supposed structural transition is lacking and poorly informative.

The Authors suggest that the dependence of excimer emission from temperature can be attributed to structural modifications induced by SCO. However, these modifications of the excimer geometry are not corroborated by an adequate crystallographic work. Differently from 1Ag, in the emission spectrum of 1Au the band due to the excimer is missing. A comment on this finding should be given.

In conclusion my opinion is that the crystallographic work and structural analysis are not of sufficiently accuracy to support the claimed novelty.

RESPONSE TO REVIEWERS' COMMENTS

Reviewer #1 (Remarks to the Author):

The authors reported two interesting catenated metal-organic frameworks in the spin-crossover field. The spin-dependent synergistic switching of dielectricity and fluorescence emission were reported. The diamagnetic zinc complex was also synthesized to compare the spin-crossover iron complex to illustrate the spin-dependent dielectricity and fluorescence properties. The time-dependent density functional theory calculations were further performed to illustrate the absorption spectra. These complexes were well characterized. However, some problems should be fixed before publication.

Response: Thank you for your kind recommendation. We have reorganized our manuscript and performed additional experiments. We hope that you will find the revised manuscript suitable for publication in Nature Communications.

1. The authors should check the address for all authors. For example, Prof. Jun Tao should belong to Beijing Institute of Technology.

Response: Thank you for pointing out the typos in our manuscript. We have corrected the address and checked the address for all authors.

2. Line 116 and 117: "Each FeII unit contains three chloroform molecules embedded within the hole of the framework, which is consistent with thermogravimetric analyses (TGA) (supplementary Figs. 5 and 6)." The chloroform has a low boiling point of 61.2°. However, the TG data show complex 1Ag can be stabilized to ~150°. It is possible that the chloroform molecules had already escaped from the sample before the TG measurement. Similarly, the TG of 1Au in Figure S11 also has the same problem. Complex 1Au can be stabilized to ~175°, which excludes the presence of methanol molecules.

Response: Thank you for your careful review and helpful comments. We collected single-crystal data for 1Ag after the crystal was kept for several months in air. Structural analyses showed that, as in the fresh sample, each Fe unit corresponds to three chloroform molecules, indicating that the single crystals are very stable. The TG data showed that the removal of chloroform occurred at an abnormally high temperature compared to the boiling point of chloroform. This high thermal stability may be due to the strong confinement of the chloroform molecules in the MOF, and a similar situation was also reported in *Chem. Commun.*, **56**, 8619-8622 (2020). The Cl...O distances between the chloroform and the NDI group are shorter than the sum of the van der Waals radii (3.270 Å), i.e., C31-Cl2...O2(2.974(10) Å, 154.9(6)°) and C32-Cl6...O4B(3.041(11) Å, 150.1(12)°) at 100 K, which indicates that there are two groups of halogen bond non-covalent interactions. These supramolecular interactions firmly fix the chloroform molecules to make the complex thermally stable. Owing to the original poor single-crystal data of 1Au, the solvent molecules cannot be resolved from the Fourier map. Therefore, we re-collected single-crystal X-ray diffraction data for 1Au on a XtaLAB Synergy R diffractometer using Rigaku (Cu) X-ray

Source (Rotating-anode X-ray tube). We successfully determined that each Fe^{II} unit contains two chloroform molecules and two water molecules embedded within the hole of the framework (100 K). At 100 K, the $\text{C5-H5}\cdots\text{Cl15C}$ (2.916(18) Å, 118.7(10)°) hydrogen bonds and the $\text{C17-Cl3}\cdots\text{O1D}$ (3.070(14) Å, 150.8(12)°) halogen bonds between the chloroform molecules and the framework lead to the removal of chloroform at an abnormally high temperature.

Fig. R1 (Supplementary Figs. 8 and 18) IR spectra (KBr pellets) of **1Ag** (a) and **1Au** (b). The presence of chloroform was confirmed by the strong peak for C–Cl stretching vibration at 764 cm^{-1} .

The presence of chloroform molecules can also be confirmed through infrared spectra, where the strong peak for C–Cl stretching vibration appears at 764 cm^{-1} (**Fig. R1**). Additionally, the elemental mapping images for single crystal and ground powder sample (**Fig. R2**) show that Cl is evenly distributed in the sample (elemental mapping photographs for other elements have also been added in **Supplementary Figs. 9-10** and **19-20**), which implies the presence of solvent chloroform molecules. Thus, we can confirm that the solvent molecules in **1Ag** and **1Au** are sufficiently stable at room temperature.

Fig. R2 (Supplementary Figs. 9-10 and 19-20) SEM image and elemental mapping photographs (Cl) of **1Ag** (a: single crystal; b: powder sample) and **1Au** (c: single crystal; d: powder sample).

We hope our explanations and revisions satisfactorily address your concerns.

3. The χ_{MT} value of **1Ag** is $4.48 \text{ cm}^3 \text{ K mol}^{-1}$, which is out of range for a HS Fe(II) ion. It is possible that complex **1Ag** contains the paramagnetic impurity. The author should carefully check the sample and make sure of the purity. Moreover, the solvents escaped from the sample as shown in TG figure. How about the sample for the magnetic measurement in Figure 3 and Figure S20. The authors should check the magnetic data with and without solvent molecules for **1Ag** and **1Au**.

Response: Thank you for your careful analysis and pointing out these issues to us. The elemental analyses and XRD results of **1Ag** are well consistent with that simulated based on single-crystal X-ray diffraction data, certifying the purity of the samples used for magnetometry.

Another source of the large χ_{MT} values might be the potential electron transfer between the redox-active Fe^{II} site and the BPND ligand, leading to the formation of Fe^{III} ions with larger magnetic moments. To exclude such a possibility, we conducted a comparative analysis of the variable-temperature infrared absorption spectra of **1Ag** and its Zn^{II}-based analogue, **2Ag**. The similarity in the absorption features of the BPND vibrational bands in both compounds, along with the absence of bands indicative of reduced BPND ligand, suggests comparable valence states for the metal ions and BPND ligands in **1Ag** and **2Ag**. This finding helps to rule out the aforementioned electron transfer as a significant contributing factor to the χ_{MT} values.

Following the diamagnetic correction in *J. Chem. Educ.* **85**, 532 (2008), we have re-calculated the Pascal's constants of **1Ag** and **1Au** in the revised manuscript. The result indicates that the original empirical diamagnetic corrections applied to our magnetic data were indeed overestimated. Notably, we re-collected the single-crystal data of **1Au** and successfully defined the solvent molecules (100 K). Therefore, the increase in molecular weight leads to an increase

in the re-calculated $\chi_M T$ value at room temperature. However, even with the corrected Pascal's constants, the $\chi_M T$ value of **1Ag** was found to be $4.08 \text{ cm}^3 \text{ K mol}^{-1}$ at 300 K, which is still significantly larger than the spin-only value of $3.0 \text{ cm}^3 \text{ K mol}^{-1}$ for a HS Fe^{II} ion. To further elucidate the origin of it, the magnetic susceptibility was calculated by the well-established *ab initio* protocol using the relativistic CASSCF/NEVPT2 (Complete-Active-Space Self-Consistent-Field/N-Electron Perturbation Theory) methods based on the truncated model extracted from the single-crystal structure of **1Ag** (250 K, HS, Supplementary Table 6). The results indicate that **1Ag** exhibits remarkable unquenched orbital angular momentum with a $\chi_M T$ value of $3.95 \text{ cm}^3 \text{ K mol}^{-1}$ at 300 K, consistent with the experimental values (Fig. R3). Notably, the slight increase in the calculated $\chi_M T$ values upon cooling was also consistent with the experimental results (200–300 K).

Fig. R3 (Supplementary Figs. 24 and 27) Comparison of variable-temperature IR absorption spectra between **1Ag** and **2Ag** (a). The experimental $\chi_M T$ curves of **1Ag** and the $\chi_M T$ values obtained from *ab initio* calculations (b).

As we have shown above, the solvent molecules in the framework are sufficiently stable at room temperature. We have attempted to prepare the desolvated samples for magnetic investigations. On the basis of the TGA results of **1Ag**, we dried the sample in vacuum at 175°C for 4 h. The TGA, XRD, and IR results (labeled **1Ag-175°C (4h)**) indicate that there are still partial solvent molecules not removed. Comparison of the XRD spectra before and after desolvation indicated that the drying process does not damage the framework structure, so the second plateau of the TGA curve ($\sim 220^\circ\text{C}$) should correspond to the loss of solvent chloroform molecules. The weight loss in the TGA curve for **1Ag-175°C** below 140°C may be due to the re-adsorbing water molecules from the air after some chloroform molecules have been partially removed. Unfortunately, due to the difficulty in completely removing chloroform molecules at 175°C (there may be re-adsorption, and higher temperature may cause decomposition), further research on the magnetic properties of the desolvated samples was not conducted.

Fig. R4 TGA curve (a), PXRD spectra (b), and IR spectra (c).

4. Is the solvent still in the sample under the condition of photomagnetic measurements?

Response: Thank you for your careful analysis. Powder samples were used for photomagnetic measurements, and the existence of solvent molecules after grinding can be confirmed by IR spectra and elemental mapping photographs (Figs. R1-R2). Therefore, we are certain that the solvents are still in the sample during photomagnetic measurements.

5. The assignment of HS and LS in Figure 3 is wrong. The isomer shift in Table S6 looks like inaccurate when comparing with Figure 3. The author should check the Mössbauer spectroscopic parameters.

Response: Thank you for pointing out our mistake. We have changed the assignment of HS and LS in Fig. 3 of our revised manuscript. We have also carefully checked the Mössbauer spectroscopic parameters based on your suggestion, and the original fitting data are shown in Fig. R5. The parameters in the text are correct.

Fig. R5 ^{57}Fe Mössbauer spectra of **1Ag** at room temperature (top) and 50 K (bottom), respectively.

6. The solvent problem should be also checked for the dielectric and fluorescence measurements.

Response: Thank you for your suggestions. Since the dielectric measurements use pressed powder samples, we can also confirm the presence of the solvent chloroform molecules by the C–Cl stretching vibration in the IR spectrum. In the fluorescence measurements, the elemental mapping images imply the presence of solvent chloroform molecules.

We have checked that the solvent molecules in the framework need to be removed at high temperatures, thus the samples are stable enough in dielectric and fluorescence measurements (10–300 K).

We hope our explanations and revisions can satisfactorily address your concerns.

7. The dielectric properties are extremely sensitive to the frequency. Such the dramatic frequency dependence is very weird. The authors should explain it.

Response: Thank you for your insightful comments. We have revised our manuscript to include a tentative explanation for the frequency dependence behavior.

During the SCO process, peaks in dielectric loss ($\tan \delta$) and imaginary part of the dielectric constant (ϵ'') were observed, exhibiting a shift toward higher temperature with increasing frequencies, showing the characteristics of dielectric relaxation (**Supplementary Fig. 31**). We noticed that such a phenomenon has also been observed in other reported SCO systems (e.g., *Angew. Chem. Int. Ed.* **57**, 8468 – 8472 (2018), *Angew. Chem. Int. Ed.* **61**, e202208886 (2022)). We have analyzed the relaxation time τ , which conforms well to the Arrhenius law, $\tau = \tau_0 \exp(E_a/k_B T)$ (where τ_0 is defined as the pre-exponential factor, E_a is the activation energy, T is

the temperature of the ε'' peak, and k_B is the Boltzmann constant), giving the activation energy of 18.0(6) kJ/mol and $\tau_0 = 2.6 \times 10^{-13}$ s (**1Ag, Supplementary Fig. 31c**). This low activation energy, along with the broad temperature range of the peak shift, suggests that the SCO transition likely occurs in a non-correlated manner across a wide timescale. Such findings are in agreement with the gradual SCO behavior observed through magnetometry. Given the few available dielectric relaxation data for SCO species, more examples are required to clearly elucidate the relaxation mechanism.

We hope our explanations satisfactorily address your concerns.

8. In Line 247 and 248, the author stated that the monotonic decrease of the pure ligand (Figure S24). However, only fluorescence spectra were shown in Figure S24. The temperature dependent emission intensities at 465 and 640 nm should be provided for the pure ligand.

Response: Thank you for your helpful suggestions. We have added the temperature dependent emission intensities at 465 and 640 nm for the pure ligand in **Supplementary Fig. 39**.

9. The temperature dependent emission intensities at 460 nm should be provided for the pure ligand to compare with that of 1Au.

Response: Thank you for your helpful suggestion. We have added the temperature dependent emission intensities at 460 nm of the pure ligand in comparison with that of **1Au** in **Supplementary Fig. 42**.

10. The cartesian coordinate information should be provided as a supporting information for the truncated molecular models.

Response: Thank you for your helpful suggestion. We have added the Cartesian Coordinates of the truncated molecular model for HS **1Ag** and LS **1Ag** in **Supplementary Data 1 (Excel file)**.

Thank you very much for your careful review. We hope our explanations and revisions satisfactorily address your concerns.

Reviewer #2 (Remarks to the Author):

Mechanically interlocked molecules (MIMs), such as catenanes, exhibit switchable properties and are of interest as components in a variety of stimuli-responsive materials. In this submitted manuscript, Wu et al. report in detail the synthesis, crystal structures, and functions of novel bimetallic Hoffman-type MOFs based on a bis-pyridyl naphthalene diimide (BPND). These MOFs have an interesting three-dimensional structure supported by a catenane-type structure formed by BPND ligands, which is found to undergo SCO upon heat or light. Furthermore, anomalous increases in maximum luminescence intensity and dielectric constant were detected simultaneously with the spin-state switching. Although these interesting dual-functional materials have been demonstrated, the authors would need to further discuss the following points.

(1) What is the significance of combining spin-dependent emission and dielectric switching in a single material? Are there any advantages for practical applications?

Response: Thank you for your insightful comments. Following your suggestions, we have added the research significance of combining spin-dependent emission and dielectric switching in a single material in the Introduction section of our revised manuscript.

In the past two decades, the integration of SCO with diverse physical or chemical properties in solid-state materials has emerged as a forefront research. This includes the synthesis and characterizations of materials exhibiting magnetic ordering/coupling, conductivity, fluorescence, dielectric/porous properties, and chemical sensing, all coupled with SCO behavior. In this work, we aim to combine crystal-structure-related properties into a single SCO material. The coupling between SCO and luminescence is usually attributed to the energy transfer due to spectral overlap between the luminescent group and the Fe^{II} SCO entity in the LS state. For octahedral Fe^{II} complexes, two electrons occupied anti-bonding 3d orbitals (*e_g*) during the transition from LS to HS state, resulting in the increase of coordination bond lengths and structural distortion. Consequently, changes in local electrical dipoles caused by spin transition can be expected, leading to changes in dielectric properties during spin transition. Therefore, through the SCO process, the photoluminescence and dielectric properties could be intimately coupled at the molecular levels. This provides the opportunity for a bi-channel (i.e. optical and electrical) read-out of the magnetic states of such systems. Moreover, the influence of the dielectric background on photoluminescence, through adjusting the energy levels of excited states with different dipole moments and also the optical properties such as refractive index, presents a unique playground for understanding the interplay between these physical properties, providing new perspectives for the development of molecular switches and multichannel devices.

We hope our explanations and revisions satisfactorily address your concerns.

(2) Correlations between catenated structures and functions

Is it possible to discuss the effect of catenane structures on physical properties more straightforwardly by comparing with similar compounds that do not have catenated structures? For example, is it possible to synthesize MOFs in which the BPNDs are not catenated by changing the synthesis conditions?

Response: Thank you for your insightful comments. We agree with you that more studies would be useful for understanding the correlations between catenated structures and functions. The methylene carbon atom in the ligand allows the two pyridine arms to rotate freely, permitting two different conformations (*syn* and *anti*) of BPND ligand. Many attempts to obtain different structures in our system have been unsuccessful, and we always obtained **1Ag** or **1Au** instead. This result indicates that **1Ag** or **1Au** is the thermodynamic principal product. We conjecture that during crystallization, the formation of interlocking dimers between BPND ligands and coordination reactions occur simultaneously, resulting in the formation of metal-organic

frameworks based on [2]catenane units. Therefore, we designed two other ligands with NDI groups (**L1** and **L2**, **Fig. R6**) and obtained single crystals of complexes **1** and **2** using the same synthetic method. Single crystal X-ray diffraction data for complexes **1** and **2** at 100 K show that complex **1** has a 2-fold interpenetrated 3D framework. For 2D complex **2**, two 3-pyridyl arms of the ligand are in *anti*-conformation, connecting two Fe^{II} ions at the diagonal sites in the rhombic grids, but the adjacent 2D layers are not further connected. The average Fe-N bond lengths of complexes **1** and **2** at 100 K are 2.178 and 2.183 Å, respectively, which are close to that of HS **1Ag** at 250 K (2.179 Å). This means that both complexes **1** and **2** are stabilized in the HS state throughout the temperature range, or SCO occurs at lower temperatures. The introduction of intermolecular contact (π -stacking and hydrogen bonding) and/or covalent bonds to bridge metal centers is beneficial for spreading cooperatively from one SCO center to another. We preliminarily hypothesize that the π - π stacking interactions within the catenated MOF predominantly contribute to the enhanced SCO properties of **1Ag**. Nevertheless, the occurrence of polymorphism and solvent-dependent SCO phenomena in Hofmann-type systems adds complexity to such analysis. Therefore, a conclusion regarding the influence of catenated structures on SCO behavior still awaits the synthesis and characterization of more non-catenated structures utilizing similar ligands for more detailed comparative investigations. The primary scientific contribution of our research lies in the integration of SCO units into MIM materials, introducing switchable photoluminescence and dielectric properties. While a detailed discussion on the catenated structures and their impact on SCO properties could be informative, we believe that the key conclusions and logical structure of our work remain robust without this supplement. Therefore, we have decided not to include the premature discussion in the revised manuscript. We hope, in future, that we can obtain suitable structures that will enable a more explicit elucidation of the relationship between structural and functional properties in these systems.

We hope our explanations satisfactorily address your concerns.

Fig. R6 Structures of L1 and complex 1 (a) and L2 and complex 2 (b).

Minor point

(1) Chemical structure of the ligand with naphthalene diimide core

The chemical structure of BPND (bispyridyl naphthalene diimide) drawn by software such as ChemDraw is not listed anywhere, including the main text and the Supplementary Information. I recommend that such a chemical structure be included in the appropriate place for the better understanding of the readers.

Response: Thank you for your helpful suggestion. We have added the chemical structure of BPND in **Supplementary Fig. 1**.

Thank you very much for your careful review. We hope our explanations and revisions satisfactorily address your concerns.

Reviewer #3 (Remarks to the Author):

The manuscript by Wu et al. reports on the synthesis and structural characterization of polycatenated bimetallic Fe(II)/Ag(I) and Fe(II)/Au(I) coordination networks of the Hofmann-type. The thermal and light induced SCO behaviour is also investigated. The reported materials show also spin-dependent photoluminescence and dielectric switching.

The novelty claimed by the Authors concerns the development of coordination networks showing the feature of mechanical link through catenation to be exploited in the integration of different spin-dependent functionalities. Although materials of the Hofman-type showing thermal and light-induced SCO as well as spin-dependent dielectricity and photoluminescence have been already reported in the literature, the novelty here is the achievement of an ordered organization of the chromophores in the solid state through catenation and pi-pi interactions. In general, the use of mechanical links in the solid state to control macroscopic properties is very interesting, however, in this manuscript the effect of catenation on the studied properties is not well investigated and described.

Response: Thank you for your insightful comments. In general, the introduction of intermolecular contact (π - π stacking and hydrogen bonding) and/or covalent bonds to bridge metal centers is beneficial for spreading cooperatively from one SCO center to another. Initially, we speculated that the π - π stacking formed in the catenated MOF is conducive to the occurrence of SCO behavior, and of course, the SCO behavior is also sensitive to crystal stacking, guest molecules, and so on. To accurately discuss the effect of catenation on SCO properties, we need to obtain more non-catenated structures utilizing similar ligands for more detailed comparative investigations. Unfortunately, numerous attempts to obtain the non-catenated structure (based on BPND ligand) in our system have been unsuccessful, which limits our ability in directly evaluating the correlation of the π - π stacking and SCO behavior. The primary purpose and contribution of our research lies in the integration of SCO units into MIM materials, introducing switchable photoluminescence and dielectric properties. We hope to obtain more suitable structures that will allow for a more detailed elucidation of the relationship between catenated structural and functional properties in the near future.

We hope our explanations satisfactorily address your concerns.

The structures are described as catena-MOF in which interpenetration occurs between 2D layers. However, the entanglement shown by the present structures is better described as polycatenation instead of interpenetration. The difference relates on whether there is or not an increase of dimensionality of the entangled array: in polycatenation there is such an increase, e.g. from 2D to 3D, in interpenetration the dimensionality of the single coordination motif is preserved e.g. from 2D to 2D (see ref. 17 of this manuscript and Carlucci et al. *Coord. Chem. Rev.* 2003, 246, 247). The analysis of the entanglement in the structures by the program ToposPro (Blatov et al. *Cryst. Growth Des.* 2014, 14, 3576) reveal that the layers have the 6-connected hxl topology. Moreover, an interesting aspect overlooked by the Authors is that among 2D layers of hxl topology no polycatenated examples have been found according to what reported in TopCryst database of

ToposPro. These aspects related to the topology and entanglement of the reported structures could be added to the manuscript.

Response: Thank you very much for the insightful suggestions and pointing out these issues to us. We agree that "polycatenation" is more accurate in our case. Accordingly, we have replaced "interpenetrated" with "polycatenated" in the structural description section. As you pointed out, we neglected the structural analysis related to topology and entanglement. According to your suggestions, we have carried out the topological analysis using ToposPro software, and the results show that the resultant topological network of **1Ag** could be identified as a 2-nodal (2, 6)-connected net with the point symbol $\{6^6. 8^6. 10^3\}\{6\}_3$ and point symbol with loops $\{3^6. 4^6. 5^3\}\{3\}_3$ (**Supplementary Fig. 12**). It is noteworthy that **1Ag** is a new topological network. Moreover, as you kindly pointed out, such polycatenated structures in 2D layers with hxl topology are unprecedented according to reports in the TopCryst database of ToposPro. All the above discussions have been added in the main text and Supporting Information.

We hope our explanations and revisions satisfactorily address your concerns.

A point of weakness of the manuscript concerns the structural analysis. Geometrical parameters such as e.g. Fe-N distances, pi-pi stacking distances and N-Fe-N angles are given to support spin-transition at different temperatures. However, no standard deviations for both single and average values are reported and, given the quality of the crystal structures, differences appear to be not very significant.

Response: Thank you for your helpful suggestion and pointing out these issues to us. We have included the standard deviations for both single and average values in the revised manuscript. In general, it is recognized that the average Fe-N bond length can contract by 0.2 Å during a complete transition from HS-Fe^{II} ions to LS-Fe^{II} ions. The magnetic data showed that about 62% (53% for **1Au**) of the Fe^{II} ions in **1Ag** underwent SCO. Correspondingly, the average Fe-N bond lengths are 2.179(7) Å (250 K) and 2.082(10) Å (100 K) for **1Ag**, and 2.178(8) Å (220 K) and 2.081(11) Å (100 K) for **1Au**, and the bond length reduction of ca. 0.1 Å corresponds to an incomplete SCO behavior in both complexes.

We hope our explanations and revisions satisfactorily address your concerns.

There are problems with the structure of **1Au** at 220 K. There are Alerts of type A in the CheckCIF file concerning the refinement of the atomic displacement parameters (ADP) of the gold atoms. The apparent impossibility to refine a reasonable ADP for such a heavy atom cast doubts on the proposed structural model. In any case, the description of the supposed structural transition is lacking and poorly informative.

Response: Thank you for your careful analysis. We have re-collected single-crystal X-ray diffraction data for **1Au** on a XtaLAB Synergy R diffractometer using Rigaku (Cu) X-ray Source (Rotating-anode X-ray tube) to eliminate this the alert of type A. The crystal data, structure refinement parameters, and selected bond lengths for **1Au** have been listed in **Supplementary Tables 2, 4-5, and 9**, and the cif file updated in CCDC. The new crystal data show that **1Au**

crystallizes in the orthorhombic space group *Ccce* at 100 K and 220 K. The average Fe-N bond lengths of 2.178(8) Å (220 K) and 2.081(11) Å (100 K) correspond to an incomplete SCO behavior in **1Au**, consistent with about 53% of the Fe^{II} ions undergoing SCO.

We hope our explanations and revisions satisfactorily address your concerns.

The Authors suggest that the dependence of excimer emission from temperature can be attributed to structural modifications induced by SCO. However, these modifications of the excimer geometry are not corroborated by an adequate crystallographic work. Differently from **1Ag**, in the emission spectrum of **1Au** the band due to the excimer is missing. A comment on this finding should be given.

Response: Thank you for your insightful comments. In general, the formation of excimer can most frequently happen when the necessary reduction of the intermolecular interplanar distance and the necessary molecular alignment are satisfied. Therefore, the regulation of spin state transition towards excimer emission is reflected in the change of spectral overlap and/or the increase of intermolecular interplanar NDI-NDI dimerization distances (3.470(18) Å at 100 K and 3.504(13) Å at 250 K) caused by the lattice expansion for **1Ag**. The small change in the interplane NDI-NDI distance gave rise to the small change in excimer emission intensity, which is not as significant as that of monomer in **1Ag**. To avoid confusion, we have rephrased the original sentence in the revised manuscript. As you indicated, no excimer emission was observed in **1Au**, which is possibly related to the presence of some non-radiative transition pathways. However, unfortunately, we could not figure out the specific mechanism responsible for the quenching of the excimer emission of **1Au** at the present stage.

Thank you very much for your careful review. We hope our explanations and revisions satisfactorily address your concerns.

REVIEWER COMMENTS

Reviewer #1 (Remarks to the Author):

This manuscript deserves to be published in Nat. Commun. as it is.

Reviewer #2 (Remarks to the Author):

The revised introduction enhances the reader's understanding of the significance of this research. Additionally, I appreciate the authors' effort to confirm the effect of the catenated structure through additional experiments. Although it is unfortunate that the effect of catenated structure on the functions could not be confirmed experimentally, I hope that further experimental results in the near future will provide insight into the relationship between mechanical entanglement and physical properties. I would recommend the revised manuscript for publication in Nature Communications.

Reviewer #3 (Remarks to the Author):

The Authors provided a revised version of the manuscript containing changes to the text and figures to answer to the Reviewer's concerns. New experiments have been also added. My opinion is that the Authors have taken into account all points aroused in the first refereeing round doing all their best to improve the manuscript. Some additional comments are:

- Concerning the clathrate solvent: the provided experimental evidence undoubtedly supports the presence of the solvent. Elemental analyses also support the given formula for the two compounds. I only suggest that could be useful to calculate from the TG curves the weight loss percentage in the temperature ranges.

- Concerning the description of the topology and entanglement some corrections are still needed: Lines 122-124: the 2-connected nodes are not relevant for the description of the topology, and the present layers are of hxl type with only 6-connected nodes. The peculiarity is that, given the bent conformation of the BPND ligands the layers have a thickness which allow parallel polycatenation (which imply a parallel disposition of the layers).

Lines 125-127: The topology of the layers (hxl) is not a new topological type, as well as, the 2D→3D polycatenation, in general, is not rare in 2D networks (see ref. 16 of the present manuscript). The novelty of the present structures is that (at the best of the present knowledge) there are no examples of polycatenation for layers of hxl topology. I made a search in TopCryst database to find 64 examples of layers with hxl topology. No polycatenation is found among these structures. The novelty of the present example of polycatenation is that it involves layers with hxl topology. The text need to be changed accordingly to previous comment.

Another interesting aspect of the two structures is that, given the same topological type and entanglement type, they show different symmetry and packing sequence suggesting possible differences in the polycatenation. However, such kind of analysis is not trivial.

RESPONSE TO REVIEWERS' COMMENTS

Reviewer #1 (Remarks to the Author):

This manuscript deserves to be published in Nat. Commun. as it is.

Response: Thank you for your careful review and kind recommendation. We have made additional revisions according to other reviewers' comments. We hope that you will find the revised manuscript suitable for publication in Nature Communications.

Reviewer #2 (Remarks to the Author):

The revised introduction enhances the reader's understanding of the significance of this research. Additionally, I appreciate the authors' effort to confirm the effect of the catenated structure through additional experiments. Although it is unfortunate that the effect of catenated structure on the functions could not be confirmed experimentally, I hope that further experimental results in the near future will provide insight into the relationship between mechanical entanglement and physical properties. I would recommend the revised manuscript for publication in Nature Communications.

Response: Thank you for your careful review and kind recommendation. We will continue to explore the relationship between mechanical entanglement and physical properties. Regarding the manuscript, we have made additional revisions according to other reviewers' comments. We hope that you will find the revised manuscript suitable for publication in Nature Communications.

Reviewer #3 (Remarks to the Author):

The Authors provided a revised version of the manuscript containing changes to the text and figures to answer to the Reviewer's concerns. New experiments have been also added. My opinion is that the Authors have taken into account all points aroused in the first refereeing round doing all their best to improve the manuscript. Some additional comments are:

-Concerning the clathrate solvent: the provided experimental evidence undoubtedly supports the presence of the solvent. Elemental analyses also support the given formula for the two compounds. I only suggest that could be useful to calculate from the TG curves the weight loss percentage in the temperature ranges.

Response: Thank you for your helpful suggestion. Based on the single-crystal data of **1Ag** and **1Au**, the calculated weight loss percentage of total solvent molecules should be 30.3% (3CHCl_3) and 21.5% (2CHCl_3 and $2\text{H}_2\text{O}$), respectively. However, discrepancies were observed between the original TG data obtained from polycrystalline sample and the calculated values based on the X-ray structures for both complexes. To ensure better thermal equilibrium and sample homogeneity, this time we used well-ground powder sample instead and reduced the heating rate from $10\text{ }^\circ\text{C min}^{-1}$ to $5\text{ }^\circ\text{C min}^{-1}$ to re-perform the TG measurements. We have updated the TG data of **1Ag** and **1Au** in **Supplementary Figs 6** and **16** of our revised manuscript. We noticed the absence of a distinct plateau in the temperature-dependent weight loss profile, likely due to the simultaneous desolvation process at high temperature ($> 270\text{ }^\circ\text{C}$) and the decomposition of the framework. This phenomenon may be explained by the fact that the removal temperature of solvent molecules is much higher than the boiling point due to the existence of halogen bonds or/and hydrogen bonds between the chloroform molecules and the framework, as discussed in the crystal structure section. The weight loss below $270\text{ }^\circ\text{C}$ corresponds to the loss of two CHCl_3 molecules for **1Ag**, and 1.5 CHCl_3 and two H_2O molecules for **1Au** (**Fig. R1**). Therefore, owing to the concurrent desolvation and decomposition in the complexes, the discussion about weight loss percentage from the TG data is not feasible. Nonetheless, during the first revision, we conducted elemental analysis, IR spectra, and elemental mapping photographs to further check the solvent molecules. Based on the experimental TG data, we added the description “The desolvation process at high temperature ($> 270\text{ }^\circ\text{C}$) is accompanied by the weight loss due to decomposition.” to the figure legends of **Supplementary Figs. 6** and **16** of our revised manuscript.

Fig. R1 (Supplementary Figs. 6 and 16) Thermogravimetric analyses of **1Ag** and **1Au**.

We hope our explanations and revisions satisfactorily address your concerns.

-Concerning the description of the topology and entanglement some corrections are still needed:

Lines 122-124: the 2-connected nodes are not relevant for the description of the topology, and the present layers are of hxl type with only 6-connected nodes. The peculiarity is that, given the bent conformation of the BPND ligands the layers have a thickness which allow parallel polycatenation (which imply a parallel disposition of the layers).

Lines 125-127: The topology of the layers (hxl) is not a new topological type, as well as, the 2D \rightarrow 3D polycatenation, in general, is not rare in 2D networks (see ref. 16 of the present manuscript). The novelty of the present structures is that (at the best of the present knowledge) there are no examples of polycatenation for layers of hxl topology. I made a search in TopCryst database to find 64 examples of layers with hxl topology. No polycatenation is found among these structures. The novelty of the present example of polycatenation is that it involves layers with hxl topology. The text need to be changed accordingly to previous comment.

Response: Thank you very much for the insightful suggestions and pointing out these issues to us. Our initial topology analysis was based on the simplified 3D framework in ToposPro software, which, in hindsight, was premature. Here, we reevaluated the description of topology and structural entanglement according to your suggestions. As you kindly pointed out, for **1Ag**, the octahedral Fe^{II} ions act as the 6-connected nodes to render 2D **hxl** layers, and the bent conformation of BPND ligands avails the formation of entanglement by parallel polycatenation. Besides, we have also confirmed that, according to reports in the TopCryst database of ToposPro, no polycatenation has been found among 2D layers with **hxl** topology.

We have deleted the original inappropriate description from the main text and included the revised sentences to better reflect the topology and structural entanglement: “The resulting 6-connected structural units lead to the formation of a 2D layer with **hxl** topology” and “It is noteworthy that due to the bent conformation of BPND ligand, the layer thickness allows for the formation of entanglement by parallel polycatenation. Moreover, according to reports in the TopCryst database of ToposPro, among 2D layers of **hxl** topology no polycatenated examples have been found.”

We are truly grateful to you for your assistance in refining the structural description and in uncovering the novelty that was initially overlooked in our manuscript. We hope our explanations and revisions satisfactorily address your concerns.

Another interesting aspect of the two structures is that, given the same topological type and entanglement type, they show different symmetry and packing sequence suggesting possible differences in the polycatenation. However, such kind of analysis is not trivial.

Response: Thank you for taking note of this interesting aspect. Indeed, as you commented, such kind of analysis is not trivial. In general, the stacking pattern of molecules in crystalline materials is determined by minimization of free energy. Larger free space typically allows molecules to adjust their orientation, optimizing various interactions. In this context, lattice volume and/or the spacing between groups will be important factors affecting the packing sequence of molecules. In our system, the unit cell volume of **1Ag** is double that of **1Au**. In addition, differences in the local chemical environment (the radius of Au⁺ and Ag⁺, solvent

molecules, *etc.*), can also subtly affect the stacking pattern of molecules, leading to changes in the overall packing symmetry of the whole lattice. Therefore, even with identical topology and entanglement, the symmetry and stacking patterns of molecules sometimes differ (e.g., *Chem. Commun.*, **46**, 8427 – 8429 (2010)). However, the above discussion only provides a tentative explanation, and the determinant factors responsible for differences in symmetry and stacking patterns in this system remain elusive at current stage. Further analysis requires careful consideration of more potential possibilities.

Thank you very much for your careful review. We hope our explanations and revisions satisfactorily address your concerns.

REVIEWERS' COMMENTS

Reviewer #3 (Remarks to the Author):

The Authors provided a revised version of the manuscript with an updated topological description according to previous comments. Also other comments have been addressed with new measurements. My opinion is that the manuscript can be accepted for publication.

RESPONSE TO REVIEWERS' COMMENTS

Reviewer #3 (Remarks to the Author):

The Authors provided a revised version of the manuscript with an updated topological description according to previous comments. Also other comments have been addressed with new measurements. My opinion is that the manuscript can be accepted for publication.

Response: Thank you for your careful review and kind recommendation.